# Theoretical Concepts in Magnetobiology after 40 Years of Research

**DOI:** 10.3390/cells11020274

**Published:** 2022-01-14

**Authors:** Vladimir N. Binhi, Andrei B. Rubin

**Affiliations:** 1Prokhorov General Physics Institute of the Russian Academy of Sciences, 38 Vavilov St., 119991 Moscow, Russia; 2Faculty of Biology, Lomonosov Moscow State University, Leninskie Gory 1/12, 119234 Moscow, Russia; rubin@biophys.msu.ru

**Keywords:** biological action of a magnetic field, nonspecific effect, random effect, molecular mechanism, kT problem

## Abstract

This review contains information on the development of magnetic biology, one of the multidisciplinary areas of biophysics. The main historical facts are presented and the general observed properties of magnetobiological phenomena are listed. The unavoidable presence of nonspecific magnetobiological effects in the everyday life of a person and society is shown. Particular attention is paid to the formation of theoretical concepts in magnetobiology and the state of the art in this area of research. Some details are provided on the molecular mechanisms of the nonspecific action of a magnetic field on organisms. The prospects of magnetobiology for the near and distant future are discussed.

## 1. Introduction

The effect of a low-intensity magnetic field (MF) on organisms has long attracted researchers. The founder of magnetic navigation, W. Gilbert (1544–1603, England), in referring to Galen, Plutarch, Ptolemy, and Paracelsus, wrote in 1600, “Others say that loadstone causes mental disturbance and makes people melancholic, and often is fatal” (see in [1] Book 1, Ch. XIV), Figure 1a. The pioneer of the scientific study of the MF’s biological effects was A.F. von Middendorff (1815–1894, Russian Empire). Based on his observations of migratory birds, he wrote, “The inner magnetic sense of our ‘sailers of heaven’ takes on the function performed by a compass for ships” ([2], p. 9), Figure 1b.

The practical study of the biological effects of MFs was initially associated with medicine. It is known, for example, that as early as the 1750s, German doctors were studying the therapeutic potential of MF by applying permanent magnets to various parts of the body [3]. There is evidence in the literature that in the 19th and first half of the 20th centuries, dozens of doctors from different countries tried to use permanent magnets and artificial DC and AC MFs for physiotherapeutic purposes [4].

Research into the nature of this phenomenon began mainly in the 1960s in the USSR and USA with the advent and development of millimeter-wave electromagnetic technology. It turned out that millimeter waves could act on plants and microorganisms [5,6]. This fact looked paradoxical, since the photon energy was two orders of magnitude lower than the activation energy of chemical transformations, while the frequency selectivity of the action and low radiation power excluded the thermal origin of the effect.

In the development of these studies, the biological effects of weak, low-frequency MFs, mostly below the intensity level of the geomagnetic field (geoMF) and in the range from a few to hundreds of Hz, were observed already in 1970 and 1980s [4,7]. The question arose: Do background MFs accompanying the transmission and use of electricity affect health? In 1983, the Problem Commission of the USSR Ministry of Health “Magnetobiology and Magnetic Therapy in Medicine” was created to coordinate research on the nature of this phenomenon and the development of therapeutic methods [8]. In 1992, the US Congress approved a five-year research program—Electric and Magnetic Fields Research and Public Information Dissemination—valued at over USD 40 million [9]. Note that by the time the program was completed in 1998, the question of whether MFs affect human health had not been unambiguously resolved. Only the following decades of collective efforts of researchers around the world made it possible to establish the approximate degree of danger of the background MFs ([10], pp. 332–333) and [11,12,13,14,15,16]. It has now been reliably established that MFs orders of magnitude smaller than the geoMF are capable of causing biological effects (e.g., [17,18,19,20,21]). More than 10,000 scientific papers have been published on the effect of weak MFs on organisms. Nevertheless, such action continues to raise questions.

Today, magnetobiology distinguishes between specific—due to special receptors—and nonspecific magnetic effects [22,23]. Specific effects occur due to specialized magnetic receptors created by nature to help some animals survive, e.g., during long seasonal migratory routes. The specific effects are a magnetic sense, i.e., magnetoreception. For example, some migratory birds have a special receptor, Figure 2a, and it is combined with the visual system. The bird can “see” changes in the MF on the order of 1/1000 of the geoMF, not only to orient itself but also to navigate along the magnetic landscape of the Earth (e.g., [24]). Some migratory insects seem to have a similar ability related to vision (e.g., [25]).

However, the main interest today is associated with nonspecific magnetic effects, Figure 2b. These are not the effects of *magnetoreception*. Already 30 years ago, MFs were considered a factor acting on a person, bypassing the sense organs [28]; that is, bypassing specialized receptors. Nonspecific effects are observed in many organisms, from protozoa and fungi to insects, plants [29], fish, animals, and humans [30]. Interest in these effects is growing since MFs can change the variety of properties. In particular, gene expression changes (e.g., [31,32]). In other words, MF is one of the factors controlling protein synthesis. However, it has not yet been possible to use the power of this method of gene control, since the nature of the primary nonspecific MF sensor in the body has not yet been explained. Such a sensor has not been identified experimentally either.

From 1980 to the present, a variety of physical mechanisms and mathematical models have been proposed to explain the biological effect of weak MFs. The list could include those that take into account: biogenic magnetite and pollution with magnetic nanoparticles; thermal heating and eddy currents; magnetohydrodynamic effects and chemical kinetics with bifurcations; cyclotron, parametric, and stochastic resonances; phase transitions in magnetic fields and the effects of spin chemistry; interference of quantum states of ions and molecular groups; coherent excitations; metastable states of water, etc. [33]. To date, only a few approaches have survived and been developed, but they still remain hypotheses.

There have been several reviews of theoretical works in the field of magnetoreception over the past decades (e.g., [24,34]). However, there have been no reviews of theoretical concepts aimed at explaining *nonspecific* effects. The purpose of this work is to fill this gap.

## 2. Phenomena Not Underlying Nonspecific MF Effects

We will immediately clarify the magnetic phenomena not associated with nonspecific sensitivity. First, there is the induction of eddy currents by a pulsed MF. Usually, they are much higher than currents produced by MFs in magnetobiology of nonspecific effects. ICNIRP guidelines [16] set a magnitude of 2 mA/m${}^{2}$ as the main limitation of the current density *j* of frequency 4–1000 Hz in the human body for the general public. Noticeable effects can be assumed to arise where a current with a slightly higher density passes through an organism; let it be j=10 mA/m${}^{2}$. For example, in [35], the transcription level in HL-60 cells changed by up to 30% at a current density of 3 to 30 mA/m${}^{2}$. Using Ohm’s law j=σE and Faraday’s law of electromagnetic induction ∇×E=−dB/dt, where *E* and *B* are electric field strength and magnetic flux density, respectively, it is easy to estimate the rate of MF change in a 1-cm sample with conductivity σ=1 Sm/m, leading to the induction of a current with a density j=10 mA/m${}^{2}$. The rate is 1 T/s. For a sinusoidal 50-Hz MF to include such rapid changes, it must have an amplitude of about 20 mT. Many magnetobiological effects occur in MFs a thousand times smaller and even in constant MFs that do not induce eddy currents. This fact indicates that substantial eddy currents in the body are impossible in such cases; the heating is all the more negligible.

Second is magnetic resonance, both electron and nuclear. Magnetic resonance is the basis, e.g., of medical tomography and scientific magnetic spectrometry. Magnetic resonance cannot underlie the nonspecific effects of MFs. Special conditions are required to observe magnetic resonance: a strong static field of several T and a perpendicular varying MF with a high-frequency component associated with the static MF. The conditions for magnetobiological effects are far from these stringent requirements. Other resonance phenomena are also impossible when one conjectures the energy accumulation of a weak MF in the oscillations of atoms and molecules. In a low-intensity MF, such an accumulation, even under idealized conditions, would require the coherence of oscillations for months or more ([36], p. 366), which would be absurd.

Based on the magnetotaxis of some bacteria, possible involvement of magnetic nanoparticles in nonspecific magnetic effects in other organisms is often discussed. Indeed, biogenic magnetite particles up to tens of nm in size have been found in many organisms. Their energy in the geoMF often exceeds kT, the characteristic scale of the activation energy of chemical reactions (e.g., [37,38]), where *k* is the Boltzmann constant, and *T* is the absolute temperature. In the MF, nanoparticles undergo a torque and could squeeze nearby biophysical structures. However, concerning nonspecific effects, this idea finds no theoretical [22] or experimental confirmation. In addition, the magnetic effect is also observed in cultures prepared to contain no nanoparticles—in various bacteria and animal and plant cell cultures. For this reason, we believe that mechanisms based on the dynamics of magnetic nanoparticles cannot underlie nonspecific magnetic effects.

Spin chemistry looks into the effect of MF on the intermediate singlet-triplet (S-T) state of reacting radicals and, hence, on the reaction yield. Note that the intermediate state of radical electrons differs from the S-T states of molecular terms. The electrons of the radicals are separated in space, and the exchange interaction is weak. In a molecule, the strong exchange interaction leads—according to the Pauli exclusion principle—to a significant energy gap between the S and T terms, on the order of 1 eV for molecular oxygen. Therefore, magnetic effects for the triplet and highly active singlet oxygen begin to be observable in fields of the order of 1 T, when the magnetic energy of the molecular electron is sufficiently high.

Lastly, there is a certain order in the chaotic motion of free ions in the biological tissues in the MF. The diamagnetic moment increases due to the deviation of the ion motion under the Lorentz force—perpendicular to the velocity vector and the MF vector. However, this effect is extremely small ([36], p. 236), orders of magnitude less than the electron diamagnetism. It is also impossible as a driving cause of magnetic biological effects in a weak MF.

## 3. Plausible Molecular Mechanisms of Nonspecific MF Effects

The rejection of the above unsuitable mechanisms leads to the conclusion that the nonspecific effect of weak MFs is the influence of MFs on the chemical process in the absence of receptors, i.e., in general conditions. There are two main stages in a chemical process: (1) the convergence of reagents, often diffusive, up to a distance of the order of the lengths of chemical bonds, and (2) the act in itself of a chemical reaction—the rearrangement of electron shells. At physiological temperature, the energy imparted by magnetic forces is many orders of magnitude less than the energy of diffusion motion, see the kT problem below. For this reason, MF cannot influence process 1.

MF, however, is capable of influencing the reaction act, i.e., process 2. The influence can occur in different ways. It can involve the spin effect on the probability of rearrangement. It can also be through one of the steric factors—through the MF effect on the spatial structure of the electron wave functions of interacting molecules. Thus, if the rearrangement of electron shells under the MF is a key factor, the following scenarios are plausible:S-T transitions in the spin state of a pair of radicals $\dot{A}\dot{B}$. Such pairs are in a spin-correlated state which is magnetically sensitive. They arise as intermediates in magnetochemical reactions involving free radicals, for example:
$$\left.\begin{array}{c} \mathrm{AB}⟷{\left(\dot{A}\dot{B}\right)}^{S}\\ \mathrm{MF}↓↑\;\;\\ {\left(\dot{A}\dot{B}\right)}^{T} \end{array}\right|⟶\dot{A}+\dot{B}$$
and in many enzyme–substrate reactions [39]. The MF, while modulating the S-T conversion, thereby changes the yield of free radicals. The energy of S-T conversion—that of the flip of spins—in the geoMF is orders of magnitude less than the energy of chemical activation; MF, here, is essentially a controlling factor only.Precession (in the vector model) of single moments—for example, the spin magnetic moments of electrons. The precession becomes uneven in a variable MF or slows down in a weakened MF [40].Interference of angular quantum states or a change in the phase of angular states of rotating molecular groups and electron clouds.

All of these quantum scenarios can be examined for their theoretical validity and suitability for comparison with experimental data. Let us take a closer look at these scenarios.

Just as an electric field interacts with electric charges, an MF interacts with magnetic moments. This is the basis for any molecular mechanism of magnetic effects in organisms. The influence of the MF on the spin—an intrinsic angular momentum of a microparticle—arises since the particle also has a magnetic moment collinear with the spin. The MF, acting on the quantum state (orientation, in the vector model) of magnetic moments, changes their spin state. In biological tissue, magnetic moments are different. In addition to the electron magnetic spin moments, these are the magnetic moments of protons, those of other magnetic nuclei, and the moments of the orbital motion of electrons or charged molecular groups.

The magnetic energy (its absolute value) of the magnetic moments—the primary “targets” of the MF, initiating the biological response—is relatively small. The energy, for example, of the electron moment in the geoMF, equal to about $3\times 10^{-9}$ eV, is seven orders of magnitude lower than the height of the chemical activation barrier, from units to tens of kT. At physiological temperature, $kT\approx 3\times 10^{-2}$ eV. This discrepancy gave rise to the so-called kT problem, which forced physicists to reject magnetobiological effects for years. Indeed, at first glance, it seems surprising that MFs, with such a big mismatch, can still cause chemical changes.

The kT problem has a simple form that compares the magnetic energy of a magnetic moment, μ, in the MF of strength *H* and the characteristic thermal energy per degree of freedom, Hμ≪kT (CGS system), where *T* is the temperature of the immediate environment of the moment. The inequality allows one to weed out many hypotheses about the nature of nonspecific effects. These are hypotheses that cannot explain why a weak MF that changes the energy of magnetic moments by only a minute amount, in the order of Hμ, is capable of producing effects that overcome thermal disturbances of those magnetic moments. Note that Hμ varies relatively little when it comes to the geoMF and the magnetic moments of microparticles. Therefore, nonspecific effects should be associated with the smallness of the effective temperature—the average thermal energy of the rotational degree of freedom of a given magnetic moment, divided by *k*. This means the insignificant dissipation that occurs due to the interaction of the moments with the thermostat. However, dissipation is negligible if the magnetic effect has time to develop and mature before the thermal equilibrium is set. This takes place, e.g., in the magnetic effects in spin chemistry.

In addition to dissipation, there is another obstacle—inertia. In inertial processes, the forced change in energy or coordinate is proportional to the square of time *t*. Then, small magnetic forces, usual for magnetobiology, can impart energy kT to a microparticle in an absurdly long time—even in the absence of dissipation and in the most favorable idealized situation. Thus, inertia cancels all hypothetical mechanisms for magnetobiology that are formulated in the language of classical physics. Possibly efficient for strong MFs of the order of Tesla (e.g., [41]), they could hardly underlie the nonspecific effects of weak MFs.

There are quantum mechanisms with no inertia. They originate in the laws governing the dynamics of angular momentum and phase of the wave function of a microscopic object. The final angular velocity of free precession is proportional to the applied torque and does not depend on time explicitly. This proportionality is a consequence of the independence of the rotation energy from the direction of the angular momentum. The angular momentum direction evolves here in proportion to *t*, i.e., without inertia.

The phase of the wave function also changes in an inertialess manner. Suppose, for illustration, there is a uniaxial quantum rotator in a coaxial MF. Let its wave function be a superposition $\Psi =\psi_{+}+\psi_{-}$ of stationary states with magnetic quantum numbers m=±1. The angular modes of these states are exp(±iφ). The wave function of a stationary state has a phase factor that depends on the energy of the state ε and on time, exp(−iεt/ℏ), where *ℏ* is Planck’s constant. In the MF, the Zeeman splitting of the initially degenerate states occurs. Their energies are now equal to $\varepsilon_{\pm }=\varepsilon \pm \Delta \varepsilon$, where Δε=γℏH, and γ is the gyromagnetic ratio. Then, the probability density of the rotator to be in an angular position φ is: $$p\left(\varphi \right)={\left|\Psi \right|}^{2}={\left|\exp\left(i\varphi -\frac{i\varepsilon_{+}t}{\hbar }\right)+\exp\left(-i\varphi -\frac{i\varepsilon_{-}t}{\hbar }\right)\right|}^{2}\propto \cos^{2}\left(\varphi -\gamma Ht\right).$$

It is an interference of quantum states $\psi_{+}$ and $\psi_{-}$. It can be seen that the motion of the nodes and antinodes of the probability density is determined by the constancy of the phase φ−γHt or, if the MF depends on time, by the equation $\varphi \left(t\right)-\gamma \int_{0}^{t}H\left(t\right)dt=Const$, implying that $\dot{\varphi }=\gamma H$. The latter relation means that the rate of displacement of the preferred angular position of the rotator does not depend on time explicitly; that is, an effect without inertia takes place.

Interference phenomena lead, for example, to the existence of atomic electron *p*-orbitals—in the form of a “dumbbell”. If there are no valence bonds with the environment, the “dumbbell” rotates with an angular velocity γH. As is seen, in a hypomagnetic field H→0, rotation slows down, which increases the likelihood of a chemical reaction. In the interference mechanism of magnetoresponse [42], this effect explained the increased probability of a biochemical reaction. Quantum interference is a modern area of molecular electronics (e.g., [43]), and interference effects in MF represent a future area of research in theoretical magnetobiology.

Thus, the elimination of the dead-end hypotheses, on the one hand, and physical requirements—the absence of inertia and small effective local temperature—on the other hand dictate the search for plausible mechanisms of nonspecific response in a certain direction. The mechanisms should rely on the quantum dynamics of magnetic moments under conditions of low thermal relaxation. These are the mechanisms of MF influence on abstract [40] and proton [44] magnetic moments, as well as on the electron spin (e.g., [23,24,45,46]) and orbital [22,47] magnetic moments. The objects of these mechanisms are, respectively, single moments, spin-correlated pairs of radicals, and quantum rotations of molecular groups within proteins.

One could also mention the developing concept of the involvement of metastable states of water and proton exchange in nonspecific magnetic effects. In recent experiments, data were obtained on various spin and structural rearrangements of water in EMF [48,49,50,51,52,53].

## 4. Why Are Nonspecific Effects Difficult to Reproduce?

As defined above, nonspecific effects occur in the absence of specialized magnetic receptors. In spite of many observations, almost all nonspecific effects were observed once—in a single laboratory. Unlike specific effects, nonspecific effects occurring in one laboratory are hard to reproduce in another. For this, magnetobiology has often been criticized. The low reproducibility is due to the influence of various random factors.

First, the values of the controlled factors are never strictly constant for many reasons. Fluctuations in the MF in office and home premises reach hundreds of nT and even more near power lines and electric transport (e.g., [54]). MF inhomogeneity inside 21 studied incubators for biological research in [55] averaged about 30 and reached hundreds of μT per 10 cm, which is often overlooked in studies of magnetobiology [56], although it is known [34,57,58] that some animals are sensitive to MF changes of 15 to 30 nT. One could expect that a similarly high sensitivity exists for other controllable factors, such as temperature. Many biological processes are sensitive to changes of 0.1 °C, although such fluctuations are typical for thermostats, hence the randomness and reduced reproducibility of the measurement results.

Second, nonspecific effects in magnetobiology depend on factors of differing natures [59,60,61], not all of which are controllable. There are more than 10 physical factors alone. In addition to the MF and the electric field, there are temperature, humidity, pressure, lighting, the rate of their changes, etc. There are also chemical, physiological, and genotypic factors.

Furthermore, even if controlled factors were almost invariable and uncontrolled ones were hypothetically absent, there would be another reason for randomness—because of the strong dependence of effects on various factors. Even small changes in MF and other controllable factors can produce an observable response. These are, for example, the responses of organisms to geomagnetic disturbance (e.g., [62,63])—its amplitude is usually less than 1/100 of the quasi-stationary geomagnetic level (e.g., [64]). The artificial reproduction of a magnetic storm has confirmed that such minute variations have their direct biological effect [65]. However, researchers in biological laboratories do not usually monitor the level of geomagnetic variations.

Finally, the low reproducibility of nonspecific effects is also associated with that randomness that occurs at a micro-level. It takes place because the conversion of the MF signal into a biochemical signal depends on internal conditions. The term “biophysical MF sensor” is applied below to explain that relationship. A biophysical sensor of the MF is a molecular structure which, (a) not being a receptor, carries magnetic moments and (b) the state of which changes depending on the state of the moments. In other words, we define a biophysical MF sensor as a converter of a physical MF signal into a chemical one. For example, a pair of spin-correlated radicals is a sensor that transforms a change in the MF into a change in the yield of radical reaction products.

As is known, homeostasis is a dynamic equilibrium, or the relative constancy of internal conditions, maintained by living systems. The fact that this equilibrium is a dynamic one indicates the presence of random deviations from it. Randomness manifests itself at all time-scales and levels of organization of living matter, from molecular ones to the level of slow evolutionary processes ([66], pp. 281–283). Thermal, chemical, and biochemical fluctuations randomize the internal conditions for the functioning of many biophysical structures, including MF sensors. Internal conditions are the physicochemical state of the local environment of the MF sensors. Internal conditions include many characteristics. These are concentrations of protein molecules and their activities and steric properties. These are also features of their interaction with a thermal bath, their rotations [67], etc.—all of them being randomly variable. Such conditions, for example, for a spin-correlated pair of radicals as a sensor, are the viscosity of the medium, details of interactions that determine the thermal spin relaxation time, the rate of the chemical reaction of radicals, concentrations, etc.

The properties of biophysical MF sensors depend on the local environment. Nonlinear changes in the parameters of sensors occur even if the primary targets of the MF are magnetic moments of the same type, for example, the moments of electrons [67]. In other words, the sensitivity of sensors to MF depends both on the magnitude of the magnetic signal and on internal conditions. Under some local conditions, a group of sensors associated with biophysical structures of one kind will respond to changes in the MF and only to changes of this magnitude. Another group of sensors in biophysical structures of a different kind will be sensitive under other internal conditions or MF magnitudes. Yet another group will be responsive in other differing conditions, etc. It follows that the activation or deactivation of distinct groups of sensors manifests itself ambiguously in the measured biological values, which leads to randomness and a decrease in reproducibility.

Through many receptor systems and feedbacks in the body, internal conditions change under biorhythms and external factors. Then, the response to a change in MF is random also due to fluctuations in internal conditions caused by the rhythms and external factors. Fluctuations in conditions constantly turn on and off various groups of biophysical sensors. Consequently, the measurement results obtained with different groups of activated sensors form not just a random variable but a set of realizations of dissimilar random variables, i.e., a heterogeneous random effect.

We add that, unlike specialized receptors located in certain parts of the body, the magnetic moments of electrons and nuclei, spin and orbital, are present in almost all biological molecules. Is it possible to say in advance which of them will transmit the MF signal to the level of biochemical reactions and where this will happen in the body? Not yet. It turns out that, despite possibly the same measured quantity, the effects arising under different internal conditions—when changing the MF, for example—are qualitatively different. They differ from each other in the same way as, for example, the effect observed in a behavioral characteristic differs from that in a protein concentration. Under these conditions—with significant heterogeneity of responses—the concepts of threshold, dose, and the MF dependence in general, can lose their meaning, which indicates the need for special methods in studying nonspecific effects.

Randomness, non-linearity, and sensitivity to many factors are inconvenient for scientific research. Nonspecific effects do not fit the “fixed effect” statistical model when the set of measurements represents a sample from same statistical population, and the exposure factor leads to a shift in the mean and/or variance. A series of measurements of the nonspecific effect contains not only the scatter of the data around the mean but also a random component in the mean value itself. This component requires special processing according to the model of “random effects” [68], opposed, in statistics, to a fixed rather than deterministic effect.

## 5. Can the Sensitivity of Organisms to Tens of nT Be Explained?

As far as the supposed mechanism of magnetic navigation in animals is concerned, it is rather “yes.” A separate visual photoreceptor is a “pixel” in the retina of an eye—a “rod” or a “cone,” well-known from school courses—which contains a protein called cryptochrome, with MF-sensitive pairs of radicals. Over billions of years of evolution, the photoreceptors of some animals have become involved in magnetoreception [24]. Many details are still unclear, but the fact itself has been confirmed by many publications (e.g., [69]).

An explanation of the MF effect on photosynthetic bacteria was proposed in [70] back in 1977. The triplet state population of a pair of photoinduced radicals in the reaction center depends on the magnitude of the MF, and this affects photosynthesis. A year later, the authors of [71] explained the biological compass—the ability of some animals to orient themselves in the geoMF—by a quantum chemical effect. They have shown that S-T conversion in radicals—taking into account the anisotropic hyperfine interaction—also depends on the direction of the MF, which is consistent with the experiment [17]. When applied to magnetic bioeffects, such mechanisms are collectively referred to as the Radical Pair Mechanism. The mechanism is known to have low sensitivity. In a separate radical pair, the geoMF causes a magnetic effect that is unlikely to exceed 0.1% of baseline chemical yield theoretically and orders of magnitude less experimentally [67]. However, the cryptochromes in the retina are arranged in an orderly fashion. The radical pairs transduce an MF change into chemical signals that are transmitted to nerves coherently, each to about 1 in $n∼10^{6}$ nerves in the bird’s eye. The compound eye of the monarch butterfly numbers up to several tens of thousands of ommatidia ([72], p. 152), each of which contains several innervated photoreceptors. The brain, as compared to the single receptor, receives about a 1000-fold ($\sqrt{n}$) signal to noise increase in magnetic sensitivity. It is a particular case of a transdisciplinary mechanism based on the central limit theorem of mathematical statistics [73,74].

The large size of the photoreceptor arrays compensates for the low sensitivity of the single photoreceptor, which is sufficient to explain high magnetic sensitivity. Thus, the magnetic receptor in these migratory species is not a separate photoreceptor but the entire set of photoreceptors together with the brain; that is, the visual analyzer as a whole.

Why is this mechanism not suitable for explaining nonspecific responses? The answer is that nonspecific effects, for example, in plants and bacteria, occur in the absence of an evolutionarily conserved receptor system associated with cognitive functions in the brain. Other properties of the mechanism, for example, insensitivity to the reversal of the MF direction [71] and lack of frequency selectivity, also do not correspond to experiments on nonspecific effects.

A single sensor must respond to a change in the MF sign and have a high sensitivity by itself to explain such effects. This circumstance indicates the direction of the search for a suitable mechanism.

First, why is the Radical Pair Mechanism insensitive to the MF reversal? Because the MF alters the dynamics of a pair of moments relative to each other. This fact means that another scenario makes sense where the MF would change the dynamics of a single magnetic moment relative to a preferred direction given by the local molecular environment [67]. Second, the low sensitivity of the Radical Pair Mechanism is due to the short lifetime of the correlated state of spins—$10^{-9}$ s, rarely $10^{-7}$ s. This lifetime is the thermal relaxation time of the electron spins (unless the chemical process involved is too fast). The relaxation time should be long enough for the MF to noticeably change the state of the spins relative to each other. However, this is quite far from the actual state of affairs in biological systems operating at temperatures of about 300 K. As a result, only relatively large MFs can produce meaningful changes.

A characteristic variable controlling the magnitude of the MF, starting from which magnetic effects become noticeable, is 1/γτ [40], where τ is the thermal relaxation time of the moment. For electrons with $\tau =10^{-9}$ s (for example, for radicals in enzyme-substrate complexes), this value is 5 mT, which is 100 times more than the geoMF. For a sensitivity of at least 5 μT often observed in nonspecific effects, the relaxation time should exceed 1 μs! It is not yet clear whether conditions could exist in living tissue, providing such a weak connection of an electron with a thermostat [75], although there are other opinions [76].

The scenario where the MF changes the dynamics of a single moment relative to its local molecular structure has attractive properties. It can explain, so far qualitatively, the features of nonspecific effects and offers numerical relationships for verification. The mechanism is maximally abstract and general so that there cannot be any alternatives among single-particle quantum ones. Its conclusions do not depend on the nature of the magnetic moments and are verifiable experimentally. Based on this mechanism, it is easy to design a scheme of experiments to reveal the nature of the biophysical sensor of nonspecific effects.

It is well-known that the elimination of the geoMF causes a variety of biological responses; see, e.g., recent experimental studies [77,78,79,80,81] and reviews [22,82]. In [40,83], it has been shown that significant changes in the dynamics of moments occur in the “zero,” or hypomagnetic, MF, when the gap ℏγH between the Zeeman sublevels becomes comparable to the width ℏ/τ of the levels themselves. As seen in Figure 3a, the sensor sensitivity to MF, i.e., the slope of the curve, nonlinearly depends on the magnitude of the field [40]. In small and large MFs, the sensor is inactive. Sensitivity occurs where γHτ∼1. Parameter γHτ controls the probability ΔP of the sensor’s response to MF. This makes it possible to find the product $\gamma \tau =1/H^{*}$ from the value $H^{*}$ of the function inflection obtained experimentally, which, in turn, allows one to clarify the nature of the primary MF target. Indeed, there are few possible targets—an electron, a proton, a magnetic nucleus, the orbital angular momentum of an electron, or a charged molecular group. At the same time, their γ and τ are significantly different and are often known in their order of magnitude for different molecular environments.

In addition, some of the primary MF sensors reside on rotating molecules [67]. According to the available literature data, responses to a weak constant MF are well pronounced in those organisms in which the processes of gene expression are most active (e.g., [27]), while nonthermal *rotations* of RNA, DNA, enzymes, synthesized proteins, etc., accompany gene expression.

As assumed in [67], an entire molecule carrying the magnetic sensor rotates. If the functioning of a magnetic sensor involves a quantum single-particle mechanism rather than the RPM, the sensor’s responsivity is dependent on the rotation speed. This dependency is due to the overlap of the two rotations. One rotation is the precession of the sensor’s magnetic moment, and the other rotation is that of the sensor itself, which rotates together with its parent molecule. To illustrate the processes, let the precession axis—that of the MF vector—and the rotation axis be collinear. When the precession rate coincides with the rotation speed, the “arrow” of the magnetic moment is at rest relative to the sensor body. This fixed magnetic moment causes the subsequent transduction of the magnetic signal to the level of biochemical reactions. The relative motion of the magnetic moment and the sensor body resembles a roulette wheel. The ball falls in the cell—this is an “effect”—when the angular velocities of the ball and the wheel coincide.

Calculations in [67] show that in a more detailed picture, which considers a gradual change in the MF vector from parallel to antiparallel through zero, the MF response of a magnetic sensor, rotating with its parent molecule, shifts from H=0. In this representation, unequal responses to the MFs of opposite directions look like a shift in response peak relative to zero, Figure 3b, which does not happen in the Radical Pair Mechanism.

In this case, the MF $H^{*}$, now corresponding to the maximum of the response, is linked to the molecular rotation rate Λ by relation $|\gamma H^{*}-\Lambda |\tau ≲1$. Thus, the experimental determination of the MF dependence of a biological measure can provide information on the nature of the molecular processes of the nonspecific response of organisms to a weak MF.

## 6. Magnetobiological Phenomena in Social Life

MFs, which penetrate the body almost unhindered, consist of various components. These are a natural geoMF and its variations and an artificial MF—an epiphenomenon of human activity. The artificial MF appeared about 200 years ago with the practical use of electricity. Today, the background level of 50/60-Hz MF in residential premises is about 10–100 nT [84] and can be several orders of magnitude higher in industrial conditions.

The spectrum of an artificial MF, Figure 4b, contains discrete components and occupies a range mainly from thousandths or less of a Hz to several hundred Hz (e.g., [54]). The nature of the continuous component is associated with randomly turning various electrical devices on and off. The random pulses of rapidly changing electric currents generate propagating electromagnetic waves. By superimposing on each other, the waves form low-frequency background electromagnetic radiation with a continuous spectrum. The intensity of the background electromagnetic field is orders of magnitude higher than that of the natural geomagnetic fluctuations, Figure 4.

Several epidemiological studies show a link between artificial MFs and health [10,11]. Various electromagnetic safety standards regulate the permissible levels of natural and artificial MFs. Some of them refer to the MF in the range that corresponds to magnetobiological effects, Figure 5. Today, these standards take into account only the absorption of electromagnetic energy, i.e., thermal effects, and the induction of eddy currents—electrochemical effects. The existence of nonthermal magnetic nonspecific effects is not reflected in the standards. The standards also differ significantly from each other and therefore remain imperfect.

The development of improved standards requires a theoretical explanation of the physical nature of the nonspecific effects. However, this is not yet available. The results of experiments, which are, in essence, an extensive bank of scattered information, have not yet been structured and theoretically generalized by any consistent theory. The current impossibility of the magnetic exposures targeted on specific biochemical processes restricts the use of magnetobiological effects in medical, pharmacological, and agricultural practices. This impossibility is, again, due to the lack of a theoretical explanation.

This situation will change in the future. However, the intensity of exploratory research is already increasing. The use of MFs is growing in therapeutic medicine [20,21,85] and for pre-sowing seed treatment of agricultural vegetables [86,87,88]. The molecular effect of magnetic therapy devices and those for the magnetic treatment of crop seeds, in which the MF exceeds several mT, is most likely associated with induction currents or reactions of spin-correlated radical pairs. So far, no such reactions have been reliably identified, although reactions involving cryptochromes are being discussed [89]. Magnetotherapy continues to be a branch of alternative medicine, in spite of the growing volume of its commercial services reaching hundreds of millions of dollars per year [90].

If the biological impact of MF were analogous to the action of chemicals, then a change in the MF magnitude or frequency would be similar to a change in the chemical composition. A change in the characteristics of the treatment—either magnetic or chemical—would alter the molecular target of the action. It is likely that by selecting certain combinations of the magnitude, frequency, and other physical factors, one could direct the MF action at one or another type of biophysical sensors in organisms. Rapid progress in the detection of genes associated with the presence of magnetic response [32,91,92] gives hope for success in this decade. What opportunities await us when the molecular mechanism of nonspecific effects becomes known? The main thing is that it will be possible to control the expression of separate genes by adjusting the MF parameters. The prospects for such an additional factor in driving molecular processes in medicine, pharmacology, and food production look attractive and promising.

To conclude, in contrast to the action of weak MFs on the receptors of some organisms, nonspecific magnetic effects are hard to detect due to their random nature. They exist inevitably and latently in all organisms, always and everywhere. However, their physical nature remains largely unknown and awaits researchers. 

## Figures and Tables

**Figure 1 cells-11-00274-f001:**
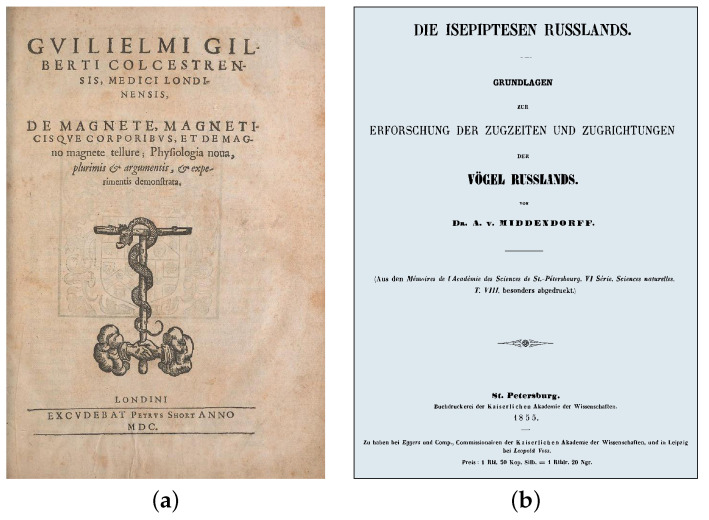
The title pages of (**a**) the first edition of W. Gilbert’s monograph and (**b**) the monograph by A.F. Middendorff with materials from a trip to Siberia.

**Figure 2 cells-11-00274-f002:**
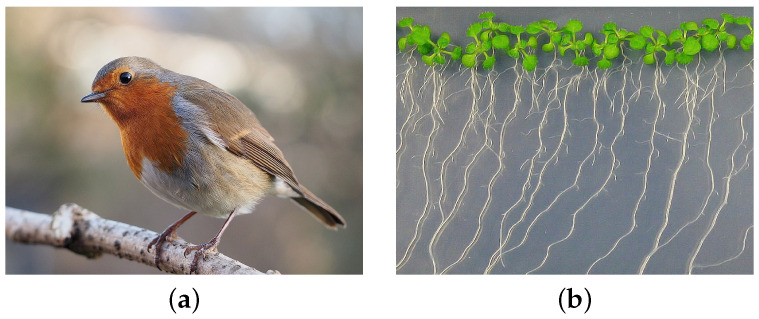
(**a**) European robin *Erithacus rubecula*, which is assumed to use a quantum magnetochemical compass for orientation and navigation in the geoMF during seasonal flights up to 2000 km [17,26]. Photo ©2014 by F.C. Franklin/CC-BY-SA-3.0. (**b**) Sprouts of the thale cress *Arabidopsis thaliana* that respond to a weak MF do not have special magnetic receptors (e.g., [27]). Photo © 2014 by Alena Kravchenko/CC-BY-SA-4.0.

**Figure 3 cells-11-00274-f003:**
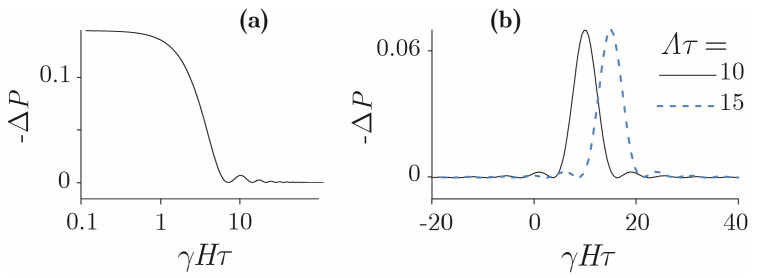
MF dependences of the probability ΔP of the primary reaction calculated for a stationary (**a**) and rotating (**b**) biophysical sensor.

**Figure 4 cells-11-00274-f004:**
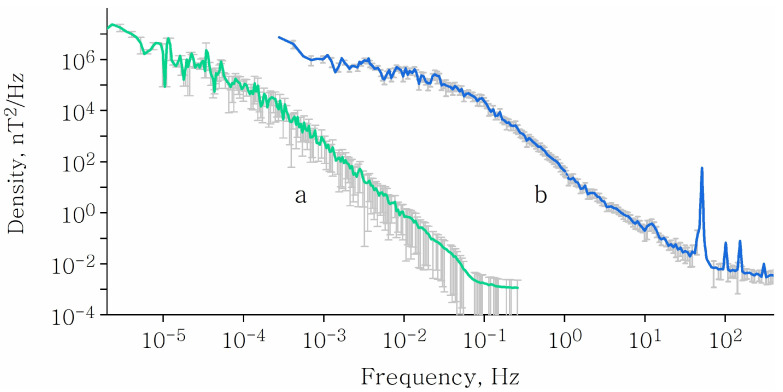
Power spectral density of the MF variations of the natural geoMF (**a**) and the urban (**b**) origin. Adapted with permission from [36], copyright 2011 © V.N. Binhi. The technical details are presented in [54].

**Figure 5 cells-11-00274-f005:**
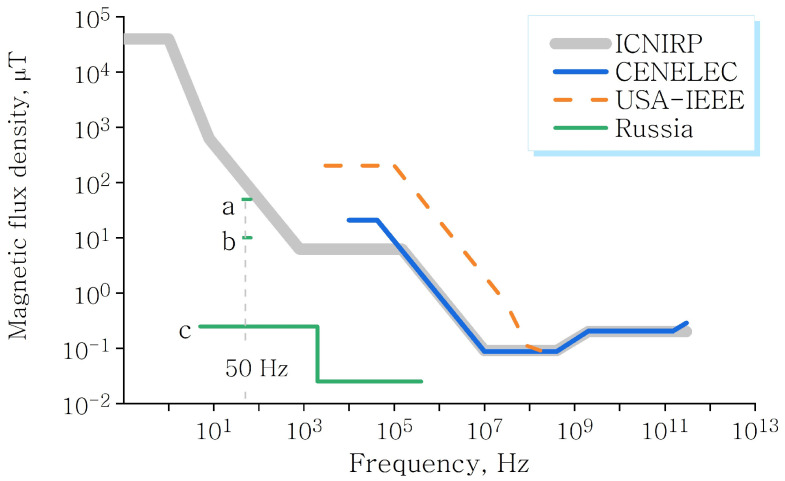
Maximum permissible levels of the EMF magnetic component for the population, adopted by various international organizations and in Russia. Russian standards: SanPiN 2.2.2/2.4.1340-03 and SanPiN 2.1.2.1002, (**a**)—residential area, (**b**)—living quarters, (**c**)—temporary standard for EMF of personal computers.

## Data Availability

Not applicable.

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
