# Peer review of "Theoretical Concepts in Magnetobiology after 40 Years of Research"

_cells, 2022, doi:10.3390/cells11020274_

Round 1
Reviewer 1 Report
Comments on
Binhi and Rubin „Theoretical Concepts in Magnetobiology after 40 Years of Research“
General:
The article reads very well, the quality of English is high. The line of argumentation is convincing. The physics and physical chemistry arguments seem to be sound, values are all in the range where I would expect them. The discussion appears to be very fair and scientific in an area which is very susceptible to esoteric and pseudo-science. I think the authors did a great job to review facts on the influence of MF on a molecular level. I do not see a reason why this shouldn’t be published as it is.
Figure 3:
I do not understand the anisotropy in the graph. Why is there a difference between orientations of the field? Or does this mean field parallel/antiparallel to the rotation normal (like torque)? To me it is not entirely clear what is plotted here? What is rotating and why, electrons or the entire molecule? If the latter how can there be a relation to an external MF. Or is this just a theoretical idealization? A sentence or two would be helpful.
May I draw the focus of the authors to a very recent article about quantum effects in magnetosensing in butterflies:
Wan et al. „Cryptochrome 1 mediates light-dependent
inclination magnetosensing in monarch butterflies“ Nature Communications 2 (2021) 771, https://doi.org/10.1038/s41467-021-21002-z
The article appear to me of high quality and scientific standard. You might consider to include it in section 5 for instance.
Author Response
The authors are grateful to this Referee for comments. The comments are marked below by <, our replies are marked with a dot.
>Figure 3: I do not understand the anisotropy in the graph. Why is there a difference between orientations of the field? Or does this mean field parallel/antiparallel to the rotation normal (like torque)? To me it is not entirely clear what is plotted here? What is rotating and why, electrons or the entire molecule? If the latter how can there be a relation to an external MF. Or is this just a theoretical idealization? A sentence or two would be helpful.
. In the model [Binhi and Prato, 2018], it was idealized that the line supporting the magnetic field (MF) vector and the axis of rotation of the molecule carrying the magnetic sensor are parallel, while the MF vector can take different values, — positive and negative values meaning opposite directions. This means that the MF vector can change from parallel, through zero, to antiparallel with respect to the angular momentum vector of the molecule. This makes it possible to calculate the effects of MFs of different signs, i.e., the MF reversal effect. An entire molecule that carries a magnetic sensor is rotating. If the functioning of a magnetic sensor is based on a quantum single-particle mechanism, and not the mechanism of radical pairs, then this functioning depends on the speed of rotation of the molecule. As explained in the article cited above, this “resembles a roulette wheel — the ball falls in the cell when the angular velocities of the ball and of the roulette wheel coincide.” The dependence on the rotation rate is not an idealization, — this is a result of the coincidence of the precession rate of the magnetic moment and that of the rotation rate of the molecule. Quantitatively, these rates are vectors. Their “arrows” pointing in the same direction either in the opposite directions give different results. Fig. 3b shows a more detailed picture, which takes into account a gradual change in the MF vector from parallel to antiparallel through zero. In this representation, unequal responses to the MFs of opposite directions look like a shift of a response peak from zero. Several explanatory sentences have been added to the text. In addition, in Fig. 3a, the vertical axis has been changed from DeltaP to minus DeltaP, as in Fig. 3b.
>May I draw the focus of the authors to a very recent article about quantum effects in magnetosensing in butterflies: Wan et al. „Cryptochrome 1 mediates light-dependent inclination magnetosensing in monarch butterflies“ Nature Communications 2 (2021) 771, https://doi.org/10.1038/s41467-021-21002-z. The article appear to me of high quality and scientific standard. You might consider to include it in section 5 for instance.
. The reference to this article has been added to the list of references.
Reviewer 2 Report
The paper is a review on the interaction of magnetic field with living organism. The paper is unclear and a lot of statement lack of references. A lot of parts are unclear and in the text authors mix low frequency MF and earth magnetic field. The paper has to be totally reviewed and authors have to go in depth of all subject. The paper has to be reorganized and references have to be updated.
Some punctual remarks:
Line 22 Please, specify what did in 1750 with MF. And also at lines 24-25
Line 27: What is the ‘MM range’?
Line 31-31 Please, clarify the sentence ‘while the spectral nature of the action and the low radiation power excluded the thermal effect.’
Line 32 what is ‘weak low-frequency MFs’ for authors? What frequency range?
Line 32-47. In this paragraph both low frequency and constant magnetic field were discussed. Please, clarify since it is confusing. In the revision, please consider also publication of WHO, IARC and ICNIRP. The effect were largely studied and published by also these international organizations.
Line 62-70. In this paragraph a lot of phenomena involving magnetic field were listed. Please, organize them in order to be not a casual list. Some ones are related to animal, other are artificial.
Line 76-85 : Please, add reference and check in literature each sentence. E.g. electroporation require high intensity voltage pulse to permeabilize membrane. Are the authors confident that transcranial magnetic stimulation is able to electropore cells? Please, add number of electric field intensity able to induces the phenomenon and references.
Line 86-84 Please, check each information repoterd here. Why authors use ‘the magnetic resonance used in medical tomographs and scientific spectrometers is excluded’ excluded from what?
Line 98 What is kT in ‘Their energy in the geoMF often exceeds kT’
Line 95-108 In this paragraph earth magnetic field (static) and time varying magnetic field were mixed and the paragraph is unclear.
Line 112 Please, what authors refer with ‘the S and T terms’ ? If it is possible use schema to describe theory.
Lines 116-121 Add related references.
Line 123-130 Authors refer for an influence on chemical reactions. What reaction? In organism? In general? Please, specify. Add references and specify the sense of the paragraph.
Line 131 Please specify what is the ‘the reaction act 2’
Line 147 Please, clarify the sentence
Line 150 Please clarify ‘Just as an electric field interacts with charges, an MF interacts with magnetic’..’ and add references
Line 152 Please, check characteristics of MF to interact with the spin?
Line 153 Please, clarify the sentence ‘arises since the particle also has a magnetic moment directly connected with the spin.’
Line 155 Why ‘In biological tissue, magnetic moments are different.’?
Line 157-166 Please, support these sentences with references and check all the sentences
Line 167 From where the Hμ ≪ kT arrive?
Line 169 ‘The inequality allows one to weed out many hypotheses about the nature of nonspecific effects’ Please, specify the meaning of the sentence and add references.
Line 167 .178 Please, clarify the sentences of the paragraph. In some point more subject where mixed. The authors know the differences of the interactions with a static and a time varying (low frequency) magnetic field? How a static magnetic field could be related to the temperature?
Line 167-183 add references. Line 183 What is T at this line?
Line 192 Please clarify what represent in line equation
Lines 190-211 Please, clarify the subject of these paragraph in relation of the topic.
Line 225 What is a ‘nonspecific effects’?
Line 232 Please, clarify the sentence ‘The inhomogeneity of the MF in biological thermostats reaches tens of μT’
Line 261-263 Please, clarify the sentences ‘Randomness — against the background of homeostasis — manifests itself….’ And ‘Thermal, chemical, and biochemical…..’
Line 267 Please, clarify the sentence ‘These are fluctuating concentrations, activities and steric restrictions of protein agents, features of interactions with a thermostat, as well as molecular rotations’
Line 268 What is ‘for a spin-correlated pair of radicals as a sensor’
Line 272-283 Authors discuss the sensors. At what they refer? Internal sensors? External? What are they?
Line 291-301 Please clarify the paragraph. It is unclear if the sensors are related to receptor. Please add references
Line 325 What is a ‘Radical Pair Mechanism’? ‘ It is known to have low sensitivity’ what?
Line 326 Please, clarify at what the authors refer with ‘photoreceptor, geoMF causes an effect that is unlikely to exceed 0.1% theoretically…..’
Line 350 ‘short life of what’? please, clarify
Line 383-393 Please, support statements with references.
Line 399 Are the authors sure that the background is 10–100 nT? In what conditions Please, check literature.
Line 403 Please, clarify the sentence ‘The nature of the continuous spectrum is associated with randomly turning various electrical devices on and off.’ Why authors analyze very low frequency under 1 Hz? Blue curve represents 50 Hz and harmonic related. Green curve what is? Noise? How these spectra were measure? Are the authors
Fig. 5 Why authors represent limit in A/m instead of T?
Lines 410-415 Please clarify sentences
Author Response
The authors are grateful to this Referee for a detailed analysis of the text and comments. The comments are marked below by <, our replies are marked with a dot.
>The paper is a review on the interaction of magnetic field with living organism. The paper is unclear and a lot of statement lack of references. A lot of parts are unclear and in the text authors mix low frequency MF and earth magnetic field. The paper has to be totally reviewed and authors have to go in depth of all subject. The paper has to be reorganized and references have to be updated.
. Apparently, the above comments summarize what the Referee addresses below. We consecutively answer each Referee’s note.
>Some punctual remarks:
>Line 22 Please, specify what did in 1750 with MF. And also at lines 24-25
. The sentence is clarified as follows, “It is known, for example, that as early as the 1750s, German doctors were studying the therapeutic potential of MF by applying permanent magnets to various parts of the body [3]. There is evidence in literature that in the 19th and first half of the 20th centuries, dozens of doctors from different countries tried to use natural magnets and artificial DC and AC MFs for physiotherapeutic purposes [4].”
>Line 27: What is the ‘MM range’?
. “MM” has been changed to “millimeter”.
>Line 31-31 Please, clarify the sentence ‘while the spectral nature of the action and the low radiation power excluded the thermal effect.’
. This phrase has been changed to “while the frequency selectivity of the action and the low radiation power excluded the thermal nature of the effect.”
>Line 32 what is ‘weak low-frequency MFs’ for authors? What frequency range?
. The phrase “biological effects of weak low-frequency MFs were discovered already in the 1970–80s” has been changed to “biological effects of weak low-frequency MFs, mostly below the geomagnetic field (geoMF) in the range from a few to hundreds of Hz, were observed already in the 1970–80s”
>Line 32-47. In this paragraph both low frequency and constant magnetic field were discussed. Please, clarify since it is confusing.
. This manuscript is about primary bio/physical mechanisms of the nonspecific biological effects caused by weak MFs. To date, there are no proved reasons why the nonspecific effects of weak low frequency MFs and those of static MFs should be explained by different primary mechanisms. For example, the Radical Pair Mechanism is not sensitive to the factor of frequency in the ELF-LF range. The Level Mixing Mechanism [Binhi and Prato, 2018, Sci. Rep.] describes effects of both DC and AC MFs within the same model. For this reason, we intentionally did not distinguish between DC and AC MFs in this manuscript.
>In the revision, please consider also publication of WHO, IARC and ICNIRP. The effect were largely studied and published by also these international organizations.
. References on the studies of epidemiology of environmental MFs have been added: Huss-ea-2018, Miller-ea-2018, Li-ea-2017, Belyaev-ea-2016, WHO-2007, ICNIRP-1998.
>Line 62-70. In this paragraph a lot of phenomena involving magnetic field were listed. Please, organize them in order to be not a casual list. Some ones are related to animal, other are artificial.
. We would like to specify that the list is that of theoretical proposed mechanisms, not of experimentally observed phenomena. If the Referee were specific as to what do he/she mean, we could reorganize the list.
>Line 76-85 : Please, add reference and check in literature each sentence. E.g. electroporation require high intensity voltage pulse to permeabilize membrane. Are the authors confident that transcranial magnetic stimulation is able to electropore cells? Please, add number of electric field intensity able to induces the phenomenon and references.
. This paragraph has been rewritten with two references.
>Line 86-84 Please, check each information repoterd here. Why authors use ‘the magnetic resonance used in medical tomographs and scientific spectrometers is excluded’ excluded from what?
. They are excluded from possible phenomena underlying the nonspecific MF effects. The sentence has been changed to “Second, it is magnetic resonance, which is the basis, e.g., of medical tomography and scientific magnetic spectrometry. Magnetic resonance cannot underlie the nonspecific effects of MF. Special conditions are required to observe magnetic resonance: a strong constant field of several T and a perpendicular varying MF with a high frequency component associated with the constant MF.” A reference has been added to the rest of the paragraph.
>Line 98 What is kT in ‘Their energy in the geoMF often exceeds kT’
. This is explained immediately after that above phrase: “exceeds kT, the characteristic scale of the activation energy of chemical reactions [e.g. 27,28], where k is the Boltzmann constant and T is the absolute temperature.” No changes are needed.
>Line 95-108 In this paragraph earth magnetic field (static) and time varying magnetic field were mixed and the paragraph is unclear.
. We would like to note that, in the context of this paragraph, the difference between DC and AC MFs is inessential. What is estimated is an approximate energy of a realistic example of magnetic nanoparticle in the body. The reference to the relevant estimates is in the paragraph.
>Line 112 Please, what authors refer with ‘the S and T terms’ ? If it is possible use schema to describe theory.
. S and T terms are the standard notation for the singlet and triplet energy levels of a molecule, as indicated at the beginning of this paragraph. In our opinion, a diagram of the energy levels for the singlet-triplet states of a molecule in a magnetic field would be redundant for this article, since such a diagram is available at the very beginning of the textbooks on spin chemistry.
>Lines 116-121 Add related references.
. Reference has been added. [Binhi-2011, p.236]
>Line 123-130 Authors refer for an influence on chemical reactions. What reaction? In organism? In general? Please, specify. Add references and specify the sense of the paragraph.
. We are talking about a chemical reaction in general, so this concept is undoubtedly applicable to chemical reactions in the body. The fact that the diffusional convergence of the reagents precedes the actual chemical reactions in solution does not require, in our opinion, citation. However, we have clarified the penultimate sentence of this section as follows, linking it to the kT problem, which is discussed in detail in several paragraphs below, «At physiological temperature, the energy of magnetic forces is many orders of magnitude less than the energy of diffusion motion, see the kT problem below.»
>Line 131 Please specify what is the ‘the reaction act 2’
. Given fragment, “the reaction act 2: in different ways,” has been changed to “the reaction act, or process 2. This can occur in different ways …” Processes 1 and 2 are explained in the preceding paragraph.
>Line 147 Please, clarify the sentence
. “All of these quantum scenarios have advantages and disadvantages that determine both their theoretical plausibility and suitability for comparison with experimental data” has been changed to “All of these quantum scenarios can be examined for their theoretical validity and suitability for comparison with experimental data”.
>Line 150 Please clarify ‘Just as an electric field interacts with charges, an MF interacts with magnetic’..’ and add references
. We would like to respectfully submit, that here we only emphasize the facts known from general physics; they do not require referencing. Is it possible to clarify the Coulomb’s law?
>Line 152 Please, check characteristics of MF to interact with the spin?
. We could not understand what the Referee meant, sorry.
>Line 153 Please, clarify the sentence ‘arises since the particle also has a magnetic moment directly connected with the spin.’
. In physics, many microparticle, while having spin, i.e., an angular momentum, have also a magnetic moment collinear with the spin. E.g., the electron magnetic moment is antiparallel to its spin momentum. The sentence now reads, “arises since the particle also has a magnetic moment collinear with the spin.”
>Line 155 Why ‘In biological tissue, magnetic moments are different.’?
. Because the substance of biological tissue contains not only electrons of free radicals, but also protons and other magnetic nuclei, as well as the magnetic moments of the orbital motion of electrons and charged molecules.
>Line 157-166 Please, support these sentences with references and check all the sentences
. All quantitative data have been verified; they are available from physics textbooks; no reference needed.
>Line 167 From where the Hμ ≪ kT arrive?
. These are two energies: the magnetic energy of magnetic moment mu in the MF of strength H and the characteristic thermal energy per degree of freedom. Fragment “The kT problem has a simple form,” has been changed to “The kT problem has a simple form that compares the magnetic energy of a magnetic moment, mu, in the MF of strength H and the characteristic thermal energy per degree of freedom,”
>Line 169 ‘The inequality allows one to weed out many hypotheses about the nature of nonspecific effects’ Please, specify the meaning of the sentence and add references.
. We have specified the meaning of the sentence by adding a sentence, “These are hypotheses that cannot explain why a weak MF that changes the energy of magnetic moments by a relatively small amount of the order of H*mu, is capable of producing effects that overcome thermal disturbances of those magnetic moments.” No references are needed.
>Line 167 .178 Please, clarify the sentences of the paragraph. In some point more subject where mixed. The authors know the differences of the interactions with a static and a time varying (low frequency) magnetic field?
. As to the “differences of the interactions with a static and a time varying (low frequency) magnetic field,” — what is it that does interact with MF? The difference could appear only with regard to some concrete mechanism of interaction. As said above, in the context of the physical mechanisms considered in the manuscript, there is no difference, see also our answer to Line 95-108 and estimates of the action of eddy currents in the very beginning of section “Phenomena that do not underlie nonspecific MF effects”.
. In addition, knowing the general reaction of a microscopic physical dynamic system — a primary MF sensor — to a sinusoidal MF, it is possible to find a solution for the particular case of a constant MF by passing to the limit to zero frequency. For this reason, there is no difference between DC and AC MF from the point of view of physical-theoretical reasoning regarding primary MF interaction with magnetic moments.
>How a static magnetic field could be related to the temperature?
. This question is explained above, following the kT problem formulation.
>Line 167-183 add references.
. No references needed; this paragraph and many others are reserved for our own grounds and inference.
>Line 183 What is T at this line?
. T here is typed in roman, not italic; so it cannot be mixed with temperature. T is the abbreviation for the unit of magnetic flux density — Tesla. Here, we have changed T to Tesla, for clarity.
>Line 192 Please clarify what represent in line equation. Lines 190-211 Please, clarify the subject of these paragraph in relation of the topic.
. This whole paragraph, line 190-211, implies that the Reader is acquainted with the basics of quantum mechanics. It would be impossible to stay within the framework of classical physics, while considering the primary mechanisms of the magnetic response. To understand these mechanisms, it would be enough to have an initial understanding of quantum physics. This review is intended for a journal of biological orientation. Therefore, the use of the terms of quantum physics is reduced to the necessary minimum. However, it would be impossible to do without these terms at all, as well as to explain such terms within this review.
>Line 225 What is a ‘nonspecific effects’?
. The notion of nonspecific effect is defined in the middle of the Introduction. “Today, magnetobiology distinguishes between specific — due to special receptors— and nonspecific magnetic effects [16,17].” And “However, the main interest today is associated with nonspecific magnetic effects. Already 30 years ago, MF was considered a factor acting on a person, bypassing the sense organs [20], that is, bypassing specialized receptors.” In other words, nonspecific effects are those that occur in the absence of specialized magnetic receptors created by nature to help some animals survive, e.g., during long seasonal migratory routes. We have inserted this brief definition in the text around line 225.
>Line 232 Please, clarify the sentence ‘The inhomogeneity of the MF in biological thermostats reaches tens of μT’
. This means that the intensity of DC or AC MFs can differ significantly in different places of a device designed to maintain a constant temperature in its internal volume. The sentence has been changed to “The inhomogeneity of the MF reaches tens of μT inside thermostats for biological research,”
>Line 261-263 Please, clarify the sentences ‘Randomness — against the background of homeostasis — manifests itself….’ And ‘Thermal, chemical, and biochemical…..’
. It is known that homeostasis is a dynamic equilibrium, or the relative constancy of internal conditions, which is maintained by living systems. The fact that this equilibrium is dynamic indicates the presence of random deviations from it. This phrase has been added to the beginning of the paragraph, line 261.
>Line 267 Please, clarify the sentence ‘These are fluctuating concentrations, activities and steric restrictions of protein agents, features of interactions with a thermostat, as well as molecular rotations.’
. The sentence has been modified, “Internal conditions include many characteristics. These are concentrations, activities, and steric properties of protein molecules. These are also features of their interaction with a thermal bath, their rotations, etc., - all of them being randomly variable.”
>Line 268 What is ‘for a spin-correlated pair of radicals as a sensor’
. “Spin-correlated pair of radicals” means that the spin states of the pair of radicals change coherently. Then, they can be described by a joint two-spin state, which is magnetically sensitive. We have clarified this in the first item of the list of mechanisms, in section “Plausible molecular mechanisms of nonspecific MF effects,” as follows. Phrase “Such pairs arise as intermediates in reactions involving free radicals, for example” has been changed to “Such pairs are in a spin-correlated state that is magnetically sensitive. They arise as intermediates in magnetochemical reactions involving free radicals, for example”.
>Line 272-283 Authors discuss the sensors. At what they refer? Internal sensors? External? What are they?
. “The biophysical MF sensor” is defined in one of the previous paragraphs: “The biophysical sensor of the MF is a molecular structure, a) which, not being a receptor, carries magnetic moments, and b) the state of which changes depending on the state of the moments.”
>Line 291-301 Please clarify the paragraph. It is unclear if the sensors are related to receptor. Please add references
. Please, see our reply to the previous questions. No references are needed.
>Line 325 What is a ‘Radical Pair Mechanism’? ‘ It is known to have low sensitivity’ what?
. “Radical Pair Mechanism” is a term that unites many similar mechanisms based on the MF effect on spin-correlated radicals. The sentence “It is known to have low sensitivity.” has been changed to “This mechanism is known to have low sensitivity.”
>Line 326 Please, clarify at what the authors refer with ‘photoreceptor, geoMF causes an effect that is unlikely to exceed 0.1% theoretically…..’
. Sentence “In a separate photoreceptor, geoMF causes an effect that is unlikely to exceed 0.1%” has been changed to “In a separate photoreceptor, geoMF causes a magnetic effect that is unlikely to exceed 0.1%”.
>Line 350 ‘short life of what’? please, clarify
. It was written “the short lifetime of the correlated state of spins”. We could not understand the question, sorry.
>Line 383-393 Please, support statements with references.
. The first half of this paragraph is already supported by two references. The second half (that has now been extended) displays our results, the reference to which has now been added.
>Line 399 Are the authors sure that the background is 10–100 nT? In what conditions Please, check literature.
. A reference has been added. [Syfers-2006]
>Line 403 Please, clarify the sentence ‘The nature of the continuous spectrum is associated with randomly turning various electrical devices on and off.’
. The random pulses of rapidly changing electric currents generate propagating electromagnetic waves, which, overlapping each other, form a low-frequency background electromagnetic radiation with a continuous spectrum. The intensity of the background electromagnetic field is several orders of magnitude higher than that of the natural geomagnetic fluctuations, Figure 4a. This phrase has been added.
>Why authors analyze very low frequency under 1 Hz?
. While speaking about the involvement of the background electromagnetic field in human life, it is necessary to indicate the characteristics of this field. Its intensity and spectrum that includes the extremely low frequency domain are the main characteristics. They are given in the review as Figure 4.
>Blue curve represents 50 Hz and harmonic related. Green curve what is? Noise? How these spectra were measure? Are the authors
. Green curve is the spectrum of the natural geomagnetic variations, which has now been explained in the figure caption.
>Fig. 5 Why authors represent limit in A/m instead of T?
. MF strength (A/m) has been changed to magnetic flux density (T).
>Lines 410-415 Please clarify sentences
. The sentences have been changed, for clarity, to “The development of improved standards requires a theoretical explanation of the physical nature of the nonspecific effects. \opt{However, this is not yet \opt{available. The} results of experiments, which are, in essence, an extensive} bank of scattered information, \opt{have not yet been structured and theoretically generalized by any consistent theory. The current impossibility of the magnetic exposures targeted on specific biochemical processes restricts the use of magnetobiological effects in medical, pharmacological, and agricultural practices. This impossibility is again due to the lack} of a theoretical explanation.”
Reviewer 3 Report
The authors describe the effect of magnetic field on organism, focusing on non-specific effects. The topic is very interesting because of an increased interest in an influence of magnetic fields on organism life processes as well as a potential risk associated with exposure to MFs in recent decades. Also, too little is known about mechanisms of MF influence on living organisms due to the complexity of this phenomenon as well as the degree and nature of this action. The manuscript is well designed, however some minor changes have to be done before accepting it for publication:
- The manuscript contains minor editing errors, please check it.
- Explain in detail the molecular mechanism of action of MF, please.
Author Response
We are grateful to this Referee for comments. The comments are marked below by <, our replies are marked with a dot.
>1. The manuscript contains minor editing errors, please check it.
. The text has been checked by a native English speaker.
>2. Explain in detail the molecular mechanism of action of MF, please.
. One can only talk about plausible mechanisms. These are listed in the “Plausible molecular mechanisms of nonspecific MF effects” section of the manuscript. As we explain in this manuscript, one of the mechanisms has certain advantages. In particular, it has predictive power and is experimentally testable. The required minimum information about regularities suitable for testing is set out in the last part of section “Can the sensitivity of organisms to tens of nT be explained?” A more detailed discussion of this mechanism would go beyond the requirements for biologically oriented texts. However, in the manuscript, in various places, several articles are referenced, where the mechanism is set out in detail. Another reference has now been added to the last paragraph of the above-indicated section.
Reviewer 4 Report
In this paper, the authors reviewed the research progresses in the last 40 years of magnetobiology. They classified most magnetobiological phenomena into specific and nonspefic, and particularly paid attention to the theoretical concepts of nonspecific magnetobiological effects.
I was excited to receive this paper for review. The paper could deliver some important information in determining the mechanism of nonspecific MF effects and answering the question why nonspecific effects are difficult to reproduce. Overall, it is a fascinating review.
I do have some comments/suggestions, which I would summarize below:
- Line 1-2, “This review contains information on the development of magnetic biology, one of the multidisciplinary areas of biophysics, as a scientific direction”: “one of the multidisciplinary areas” and “a scientific direction” seem redundant to me. Would it be better to delete “a scientific direction”?
- Line 27, “MM range”: The abbreviation “MM” needs a full terminology or phrase since it is the first time use in the paper.
- Figure 2 is not very informative here. I would suggest to include another picture related to nonspecific magnetobiological phenomena/mechanism as figure 2b.
- The authors discussed MF mediated S-T transitions in the spin state of a pair of radicals. Is it possible that MF could also affect electron transport or proton transport process by any means? besides of the electron spin status? Since electron transport and proton transport are essential process in biological systems.
Author Response
We are grateful to this Referee for comments. The comments are marked below by <, our replies are marked with a dot.
>Line 1-2, “This review contains information on the development of magnetic biology, one of the multidisciplinary areas of biophysics, as a scientific direction”: “one of the multidisciplinary areas” and “a scientific direction” seem redundant to me. Would it be better to delete “a scientific direction”?
. “as a scientific direction” has been removed.
>Line 27, “MM range”: The abbreviation “MM” needs a full terminology or phrase since it is the first time use in the paper.
. This abbreviation “MM” has been changed to “millimeter”.
>Figure 2 is not very informative here. I would suggest to include another picture related to nonspecific magnetobiological phenomena/mechanism as figure 2b.
. A picture related to nonspecific magnetobiological phenomena has been added as Figure 2b, with due caption.
>The authors discussed MF mediated S-T transitions in the spin state of a pair of radicals. Is it possible that MF could also affect electron transport or proton transport process by any means? besides of the electron spin status? Since electron transport and proton transport are essential process in biological systems.
. On the one hand, electron transport is implicitly present in the RPM, as a process, the probability of which depends on the spin state of the radical pair — a spin factor, and on the space peculiarities of the electron cloud — sort of a steric factor. Both factors are listed as possible mechanisms in section 3 of the manuscript. On the other hand, taken independently of spin or steric factors, the electron transport is a rearrangement of electron shells that looks like a motion of a classical particle. In this case, to me, there is no possibility for the MF to be connected to this motion other than through the Lorentz force. However, there are well-known estimates regarding the impossibility for biological magnetic effects to be explained by the Lorentz force. In addition, this process is extremely fast compared to the spin or other magnetic moments precession. For these reasons, I am skeptical regarding MF influence on the electron transport. As to the proton transport, I studied this by use of the model [Binhi, 2019, J. Chem. Phys., 151, 204101] — with a negative result for proton, only electron transport (spin-dependent transport, but more direct than in the RPM) had a more or less acceptable result. In my opinion, protons is attractive as a possible MF target, but only due to the fact that its spin has a very long lifetime.
Round 2
Reviewer 2 Report
The paper was modified but my previous questions h
Some information are doubt since in contrast with the main literature. My opinion authors have to revise again accurately the literature. Some technical information are incorrect.
Some main remarks (not exhaustive) belove.
e.g. line 96-98 ‘Many magnetobiological effects occur in MFs a thousand times smaller and even in constant MFs that do not induce eddy currents. This fact indicates that substantial eddy currents in the body impossible; the heating is all the more negligible’
In fact the main effect describe in the literature (ICNIRP documents) of AC MF are eddy currents. In DC eddy currents are induced by the movement of the body in the static MF.’
In contrast authors discuss the ability of the MF field to interact at chemical level with the matter. They state that there is a chemical effect, but not eddy currents. Nevertheless, it is well known that an electric field moves charges and generate drift currents. When a magnetic field exist also an electric field exists.
It is well known that some animals have magnetic receptors able to orient them. In my advice since there is a big quantities of literature that describes the effect in living organism the effects of electric and magnetic field it is a too strong the statement ‘This fact indicates that substantial eddy currents in the body impossible’. Moreover, ICNIRP literature well describes time varying effects of magnetic field and DC magnetic field and differences.
How the spectra in Fig. 4 were obtained? What instrument? What is the sampling time? What the extraction method? Instruments are characterized by low frequency noise related to electronic components inside. Are the authors sure that this spectra represent the information described in the text?
In Fig. 4 no components at frequencies for which ‘propagation’ phenomena is appreciable are shown. At low frequency propagation is not generally described since wavelength is comparable to the earth diameter. The harmonic components related to switching frequencies are at higher frequency with respect to switching frequency and depends on the time derivative of the switching pulse if it is represented by a square pulse (expected frequency are higher than kHz, MHz of higher)
Not all sentences are supported by literature. Also basic information, primarily if they show numbers and quantities, have to be referenced.
Line 450 a background level of magnetic field in residential environment of 10-100 nT is very difficult to be measured with available instruments. Also in this matter a lot of literature investigated residential environment an show measured field values. Check literature of from the end of 1900 until e.g. 2005.
A suggestion could be to focus the description on animal phenomena in DC field and related phenomena.
Author Response
The authors are grateful to this Referee for comments. The comments are marked below by >, our replies are marked with a dot.
>Some information are doubt since in contrast with the main literature. My opinion authors have to revise again accurately the literature. Some technical information are incorrect.
. This comment summarizes what the Referee addresses below. We consecutively answer each Referee’s note.
>line 96-98 ‘Many magnetobiological effects occur in MFs a thousand times smaller and even in constant MFs that do not induce eddy currents. This fact indicates that substantial eddy currents in the body impossible; the heating is all the more negligible’. In fact the main effect describe in the literature (ICNIRP documents) of AC MF are eddy currents. In DC eddy currents are induced by the movement of the body in the static MF.’
. As the Referee advised in the previous review, we have added a dedicated paragraph evaluating the effects of induced electric field based on the ICNIRP guidelines. As we have shown, these effects arise when the rate of the MF change exceeds 1 T/s in the order of magnitude. There are many publications reporting the observation of biological effects in variable MF with a rate of its change that is orders of magnitude lower than 1 T/s. There is obviously no reason to believe that these effects are caused by an induced electric field.
. Many laboratory experiments with cell cultures and large organisms are carried out in a constant MF and under conditions that exclude movements, in particular, turns. The effects of the induced MF are excluded here, and the observed responses should be associated with a different mechanism of MF action, — not with eddy currents. On the other hand, in the coordinate system associated with the human body, when the body turns in the geomagnetic field, the rate of change in the MF does not exceed 0.001 T/s. Because of this, we do not see sufficient grounds for changing the essence of our above quoted statement. However, for the sake of clarity, the last sentence has been changed to “This fact indicates that substantial eddy currents in the body impossible in such cases; the heating is all the more negligible”.
>In contrast authors discuss the ability of the MF field to interact at chemical level with the matter. They state that there is a chemical effect, but not eddy currents. Nevertheless, it is well known that an electric field moves charges and generate drift currents. When a magnetic field exist also an electric field exists.
. We agree that an alternating, but not constant, MF induces an electric field. But the question is what the magnitude of this field is. As we have shown in the first paragraph of the section "Phenomena not underlying nonspecific MF effects", in many observations of magnetic nonspecific effects, the induced electric field is vanishingly small.
>It is well known that some animals have magnetic receptors able to orient them. In my advice since there is a big quantities of literature that describes the effect in living organism the effects of electric and magnetic field it is a too strong the statement ‘This fact indicates that substantial eddy currents in the body impossible’. Moreover, ICNIRP literature well describes time varying effects of magnetic field and DC magnetic field and differences.
. As we explained above, this statement has been changed to “This fact indicates that substantial eddy currents in the body impossible in such cases”. This refers to cases of observation of magnetic effects in the absence of any perceptible induced electric fields.
>How the spectra in Fig. 4 were obtained? What instrument? What is the sampling time? What the extraction method? Instruments are characterized by low frequency noise related to electronic components inside. Are the authors sure that this spectra represent the information described in the text?
. Yes, we are sure that the spectra in Fig. 4 correspond to the information described in the text. Since this manuscript is a review, the technical details would not be appropriate for this type of publication. However, for the sake of clarity, we have added a sentence to the caption to Fig. 4, “The technical details are presented in [54]”.
>In Fig. 4 no components at frequencies for which ‘propagation’ phenomena is appreciable are shown. At low frequency propagation is not generally described since wavelength is comparable to the earth diameter. The harmonic components related to switching frequencies are at higher frequency with respect to switching frequency and depends on the time derivative of the switching pulse if it is represented by a square pulse (expected frequency are higher than kHz, MHz of higher)
. Fig. 4 shows the power spectra of the MF in the frequency range relevant for magnetobiology, with a certain margin. High-frequency spectra of artificial MFs and electric fields are of no interest for magnetobiology. Magnetobiology, in contrast to electromagnetobiology, is interested in the action of a separate magnetic component, when it represents a separate entity for the organism with good accuracy, i.e. in the low frequency range.
. An electric current switching pulse is usually a step or a rectangular pulse of long duration, the spectrum of which also includes the low-frequency region up to frequencies of the order of 1/tau, where tau is the pulse duration. The range of switching times for urban DC electrical devices (mainly electric transport and industry enterprises) covers a wide range from milliseconds (limited by the inductance of the lines) to days in the order of magnitude. This corresponds to the frequency range of Fig. 4. Low-frequency spectral components propagate, despite their very long wavelength. At distances of the order of 10–1000 m, they do not propagate in the form of sinusoidal waves, but in the form of a relatively smoothly increasing field strength. Therefore, there is no need to change Fig. 4 or accompanying text, with the exception of the above-mentioned added sentence to the caption.
>Not all sentences are supported by literature. Also basic information, primarily if they show numbers and quantities, have to be referenced.
. We have carefully examined the text. All numerical data are either referenced or widely known. We only add a reference to the level of geomagnetic variation 1/100 of the quasi-stationary geomagnetic level and a reference to the magnitude of the MF inhomogeneity in incubator thermostats for biology needs.
>Line 450 a background level of magnetic field in residential environment of 10-100 nT is very difficult to be measured with available instruments. Also in this matter a lot of literature investigated residential environment an show measured field values. Check literature of from the end of 1900 until e.g. 2005.
. The value of the background level of the magnetic field in residential premises has already been accompanied by a literary source in the original and R1 versions. We respectfully submit that measuring the 50-Hz magnetic field at 100 nT level with the same accuracy is available even with modern smartphones with suitable applications from Google Play, etc. 10-nT variations are easily measurable by modern laboratory devices produced by Barrington C, Stefan Mayer C, etc. Apparently, the referee meant picoTesla.
>A suggestion could be to focus the description on animal phenomena in DC field and related phenomena.
. As the manuscript title suggests, the text is devoted to a review of the theoretical mechanisms of nonspecific responses to weak MF — both DC and AC in the low-frequency region. Such fields act on biophysical magnetic sensors in the body in the same way, so there is no reason to be limited only by DC MFs. The review contains references to experimental works and facts that are essential for understanding the theoretical concepts in magnetobiology. The description of animal phenomena was not the goal of our work.